# Bleeding Complications of Anticoagulation Therapy in Clinical Practice—Epidemiology and Management: Review of the Literature

**DOI:** 10.3390/biomedicines12102242

**Published:** 2024-10-01

**Authors:** Maciej Kocjan, Michał Kosowski, Michalina Mazurkiewicz, Piotr Muzyk, Krzysztof Nowakowski, Jakub Kawecki, Beata Morawiec, Damian Kawecki

**Affiliations:** 12nd Department of Cardiology, Faculty of Medical Sciences in Zabrze, Medical University of Silesia, 40-055 Katowice, Poland; maciej.kocjan@onet.eu (M.K.); michalina.liput@gmail.com (M.M.); piotrekmuzyk1989@gmail.com (P.M.); jkawecki@sum.edu.pl (J.K.); d.kawecki@interia.pl (D.K.); 2Department of Internal Medicine and Clinical Pharmacology, Medical University of Silesia, Medyków 18, 40-752 Katowice, Poland; farmklin@sum.edu.pl; 3Department of Urology and Urological Oncology in Rybnik, Faculty of Medical Sciences in Zabrze, Academy of Silesia in Katowice, 40-055 Katowice, Poland; krznowakowski@gmail.com

**Keywords:** direct oral anticoagulants, heparins, vitamin K antagonists, fondaparinux, anticoagulation therapy, bleeding complications

## Abstract

Due to their very wide range of indications, anticoagulants are one of the most commonly used drug groups. Although these drugs are characterized by different mechanisms of action, the most common complication of their use is still bleeding episodes, the frequency of which depends largely on the clinical condition of the patient using such therapy. For this reason, to this day, the best method of preventing bleeding complications remains the assessment of bleeding risk using scales such as HAS-BLED. There are many reports in the literature assessing the occurrence of this type of complication after the use of drugs affecting the coagulation process, as well as many reports comparing individual groups of drugs with different mechanisms of action. However, there are still no clear guidelines that would indicate which group of anticoagulants should be preferred in particular groups of patients. The aim of our article is to summarize the data collected so far regarding the safety of using specific groups of anticoagulants and the frequency of bleeding complications after their use.

## 1. Introduction

Due to the very wide indications for use, anticoagulants (ACs) are nowadays used by a very wide group of patients. The most common of them are atrial fibrillation (AF), deep vein thrombosis (DVT), pulmonary embolism (PE) and ischemic stroke [1,2,3].

Because of the mechanism of action, these drugs can be divided into a few groups: vitamin K antagonists (VKAs), heparins and their derivatives, and direct oral anticoagulants (DOACs) [4].

Despite the different mechanisms of action of these drugs, the most common group of complications of their use are bleeding complications [5]. Their frequency depends largely on the clinical condition of the patient, their age, comorbidities and the drugs they use [6].

There is very often a fine line between anticoagulant effect and hemorrhagic complications. To determine the balance between benefit and risk in the best way, there are popular scoring systems related to stroke, such as CHA2DS2-VASc [7], as well as scoring systems assessing bleeding risk, such as HAS-BLED, GARFIELD-AF, ATRIA, ORBIT [8,9,10,11].

It is, however, the HAS-BLED scale that has found an important place in clinical practice, originating from the 2010 European Heart Survey database [10] and focusing mainly on modifiable bleeding risk factors. It is routinely recommended for predicting bleeding risk in patients with atrial fibrillation taking anticoagulants [7,12].

Although these scales answer the question of whether anticoagulant drugs can be safely administered, they do not provide information on which drugs to prefer in given groups of patients to reduce this risk.

The aim of our review is to discuss the available data comparing the risk of bleeding events after the use of individual groups of anticoagulants, as well as to discuss potential therapeutic methods in the event of such events.

## 2. Heparins—Mechanism of Action

Drugs belonging to the heparin group can be divided into two groups: unfractionated heparin (UFH) and low molecular weight heparins (LMWH). The mechanism of action of UFH and LMWH is not completely the same, which is largely due to differences in molecular structure.

They are composed of repeating disaccharide units (iduronic acid/glucoronic acid-glucosamine), but UFH is characterized by a larger number of these units in its structure, and therefore a higher molecular weight (12–15 kDa in case of UFH versus 3–9 kDa in case of LMWH), which means that they preferentially bind antithrombin with thrombin (clotting factor IIa). The chemical structure of the heparin molecule is presented in Figure 1.

Due to their lower molecular weight, LMWHs have a lower ability to inhibit factor IIa, but they have very strong activity against factor Xa, inhibiting the coagulation system at a higher level [15,16]. The main mechanism of action of this group of anticoagulants is shown in Figure 2.

The basic indications for the use of UFH are DVT, PE and thromboprophylaxis in AF, but this drug is also widely used in off-label indications like in the case of patients with acute coronary syndrome during percutaneous coronary intervention [17].

Among the numerous indications for the use of LMWH, the most important is the prevention of DVT, treatment of venous thrombosis, PE, myocardial infarction with ST segment elevation, unstable angina or prevention of clotting in extracorporeal circuits [18].

When considering the mechanism of action of both UFH and LMWH, it should be remembered that, apart from the basic mechanism of binding to coagulation factors through antithrombin, heparins, due to the accumulation of a negative charge on their molecules, interact very strongly with positively charged molecules found in biological membranes and plasma. This phenomenon not only makes their pharmacokinetics unstable, but also causes complications such as heparin-induced thrombocytopenia (HIT). These effects are much stronger in the case of unfractionated heparin [19].

The molecule resulting from the evolution of heparin, fondaprinux, is devoid of the phenomena discussed above. It consists only of a pentasaccharide sequence, which allows it to selectively bind to the antithrombin molecule and increase its factor Xa-inhibiting activity, without affecting thrombin [20]. Due to this mechanism of action, it has 7 times stronger anticoagulant effects compared to LMWH, while also being characterized by more stable pharmacokinetics and a lower incidence of complications [21]. It is also worth noting that, due to the same mechanism of action as in the case of LMWH, the scope of indications for its use coincides with the scope typical for heparins.

## 3. Oral Anticoagulants—Mechanism of Action

Oral anticoagulants are very commonly used in daily clinical practice. There are two groups of drugs in this category: vitamin K antagonists and novel oral anticoagulants, more commonly referred to as non-vitamin K antagonist oral anticoagulants or direct oral anticoagulants. The first group consists of two molecules: warfarin (WAF) and acenocomarole, the latter group includes dabigatran (DAB), rivaroxaban (RIV), apixaban (API), edoxaban (EDO), and betrixaban. Both groups differ in their mechanism of action.

VKAs by inhibiting the enzyme vitamin K epoxide reductase C1 (VKORC1), prevent carboxylation of vitamin K-dependent proteins K in hepatocytes, especially coagulation factors II, VII, IX and X. This leads to impairment of partially gamma-carboxylated vitamin K-dependent factors due to their inability to bind activated platelets [22]. Given the long half-life of prothrombin synthesis by the liver, i.e., 60 h, a stable anticoagulant state cannot be achieved until at least 5 days after the start of VKA therapy. The use of VKAs involves regular monitoring of the international normalized ratio (INR) (derived from prothrombin time (PT), which reflects the global hypocoagulant state with specific target values depending on the indication for VKA use. Over the years, attention has been paid to dietary recommendations by limiting the consumption of vitamin K-containing products in VKA-treated patients, to minimize interaction and drug interference [23].

DOACs have a more predictable effect by directly inhibiting the activation of clotting factors that act late in the clotting cascade. In addition, DOACs show a faster onset of action and a shorter half-life, and the most convenient features are the lack of need for regular laboratory monitoring and the absence of dietary and drug interactions [24] (Figure 3).

DOACs are recommended as the first-line treatment for most patients with AF and are indicated also in prophylaxis of DVT leading to PE in patients after a hip or knee replacement surgery, and treatment of DVT and PE to reduce the risk of recurrence. They are considered an advance over the traditional VKAs-based approach based on several advantages: safer and more effective profile than VKAs, inducing up to half as much life-threatening bleeding than VKAs, more convenient and safe use in fixed doses and also no requirement of frequent blood monitoring. Nevertheless, VKAs remain the only OAC with class I indication for the anticoagulant treatment in patients with implanted mechanical valves, with class III indication for DOACs. The following table (Table 1) summarizes the effects and pharmacokinetics of individual DOACs.

## 4. Risk of Bleeding

The widespread use of anticoagulants is mainly due to the need for thromboprophylaxis for atrial fibrillation, which is by far the most common sustained arrhythmia worldwide and is associated with an average fivefold increase in the risk of thromboembolic events [25,27]. Atrial fibrillation is diagnosed in 20–25% of ischemic stroke patients [28], and this percentage rises up to 30% among those undergoing long-term cardiac rhythm monitoring [29].

On the other hand, the prevention of central nervous system and peripheral embolism is closely associated with an increased risk of bleeding [30]. Serious bleeding in the treatment of AF is closely associated with increased risk of death and serious adverse outcomes in both the short and long term, but all bleeding is associated with reduced quality of life.

The most used definition of major bleeding in non-surgical patients according to the International Society on Thrombosis and Hemostasis (ISTH) includes:(1)fatal bleeding and/or(2)symptomatic bleeding in a critical area or organ, such as intracranial, intraspinal, intraocular, retroperitoneal, delivery or pericardial sac bleeding, or intramuscular bleeding with fascial compartment syndrome and/or(3)bleeding accompanied by a decrease in hemoglobin of ≥2 g/dL or leading to transfusion of ≥2 units of whole blood or red blood cell concentrate [31].

However, guidelines published in Circulation in 2011 suggest that a better way to describe bleeding complications that better reflects the prognosis of patients with such complications is to use the Bleeding Academic Research Consortium (BARC) scale which divides it into five types (Table 2) [32]:

The annual incidence of major bleeding ranges from 1.3% to 7.2% in patients with AF treated with VKA [30]. Because of the distinction between major and minor hemorrhagic complications, intracranial hemorrhages occupy a much more special place in terms of morbidity and mortality among major hemorrhages. Intracranial hemorrhages, which have an annual incidence of 0.2%, are reported to be about 0.5% in major DOAC trials, while this number can be as high as 0.9% in patients using VKAs [33,34,35]. Bleeding is also known to reduce quality of life and the adherence to medical recommendations for the use of anticoagulation. In this sense, bleeding risk scales are important tools to help predict serious bleeding. Serious bleeding has been evaluated as a safety endpoint in studies comparing WAF with DOACs, such as DAB, RIV, API and EDO [33,34,36].

The RE-LY trial compared DAB and WAF. Taking DAB 150 mg 2 × daily (but not DAB 110 mg 2 × daily) was associated with an increased risk of major gastrointestinal bleeding compared with WAF (RR 1.50; 95% CI 1.19–1.89) [37]. In the analysis of patients with a GFR of 30–49 mL/min, a lower risk of stroke/embolism and a similar risk of major bleeding was demonstrated with DAB compared to WAF [38]. The ROCKET-AF trial compared RIV and WAF. In elderly patients (as many as 44% were over 75 years old), the primary composite endpoint, stroke, or peripheral embolism, occurred more frequently in the elderly population (2.57% vs. 2.05%/100 patient-years; *p* = 0.0068); similar differences were observed for major bleeding (4.63% in the elderly vs. 2.74%/100 patient-years in the younger population; *p* < 0.0001) [39]. In another analysis of this study, Piccini et al. (2016) [40] focused on the risks associated with the use of RIV together with other drugs. Polypharmacy did not result in an increased risk of stroke or peripheral embolism but was associated with an increased risk of major and non-major bleeding (risk ratio [HR, hazard ratio] 1.47 when taking >10 drugs vs. group of patients taking ≤4 drugs, 95% CI 1.31–1.65). There were no differences in prognosis between patients treated with WAF and RIV, except for patients taking less than five drugs. Among these patients, RIV use was associated with a lower risk of major bleeding. The ENGAGE AF-TIMI 48 trial [41] compared EDO 60 mg and 30 mg and WAF. Both once-daily regimens of EDO were no worse than WAF in preventing stroke or systemic embolism and were associated with significantly lower rates of bleeding and cardiovascular death. In the ARISTOTLE trial [36] when comparing API to WAF, it was API that had a lower rate of major bleeding.

The above-mentioned landmark phase 3 randomized controlled trials (RCTs) demonstrated that DOACs, as recommended by the ISTH, are at least as effective as WAF in preventing stroke and systemic thromboembolism and are preferred over WAF in patients with nonvalvular AF [42,43].

In recent years, there has been no shortage of studies comparing the effectiveness and safety of DOACs with heparins. A meta-analysis of randomized clinical trials (RCTs) published in 2022 and concerning patients after total knee arthroplasty clearly showed that the use of enoxaparin is associated with lower effectiveness in terms of preventing recurrence of venous thromboembolism compared to DOACs. More important, however, is the fact that no differences were observed in the incidence of bleeding complications [44]. On the other hand, a meta-analysis conducted on a group of patients hospitalized for various reasons who received anticoagulants for the prevention of venous thromboembolism did not indicate a higher effectiveness of DOACs compared to LMWH, and the incidence of bleeding was higher in the group taking DOACs [45]. These facts lead to a conclusion, that there are still no clear data that would determine which group of anticoagulants should be preferred as safer considering their bleeding risk.

The final issue is the comparison of the safety of fondaparinux to other anticoagulants. Due to the very specific mechanism of action described above, it can be expected that its use will be associated with a lower risk of complications. Three meta-analyses from 2016, 2017 and 2019 unanimously indicated that, compared to LMWH, the use of fondaparinux is associated with a lower risk of bleeding complications [21,46,47].

### Special Patients at Higher Risk of Bleeding

Special attention should be put on specific patient populations at higher risk of bleeding during anticoagulant therapy with special focus on patients with chronic kidney or liver disease, older and frail patients, oncological patients.

Patients with chronic kidney disease (CKD) and AF are more likely to experience complications and death due to an increased risk of both thromboembolic events and severe bleeding [46]. All DOACs are excreted by kidneys, although to varying degrees, (with a maximum for DAB (80%) and disposed unchanged. Renal function should be monitored at least annually in all patients receiving DOACs. Patients with a CrCl of less than 30 mL/min were generally excluded from all available DOAC trials (except for a small group of patients with a CrCl of 25–30 mL/min taking API in the ARISTOTLE trial). RIV, API and EDO (but not DAB) are approved in Europe for use in patients with severe CKD (stage 4, i.e., CrCl 15–29 mL/min), at reduced doses. In Europe, DOACs should not be prescribed for patients with AF and severe renal dysfunction (CrCl < 15 mL/min) or for patients on dialysis [48,49]. In this particular group of patients, we should consider the use of LMWH in anticoagulation therapy, which is confirmed by numerous retrospective and registry studies, indicating that in patients with kidney damage, the use of LMWH is associated not only with a lower risk of bleeding, but also with generally lower mortality compared to with the use of DOACs [50,51].

Another group are patients with low body weight, in which the risk of death, stroke and SE and major bleeding with anticoagulation is higher than in normal-weight patients [52]. Body weight ≤ 60 kg is a criterion for reducing the dose of API [53]. In the largest study evaluating the effect of DOACs according to body weight in patients with AF, API was at least as effective as WAF but safer for any value of body weight, with the greatest reduction in the risk of bleeding and hemorrhagic stroke in the group with body weight ≤ 60 kg [52]. Similar studies were also conducted on a group of obese patients in which no significant differences have been shown between individual anticoagulants, both in terms of their effectiveness and the risk of bleeding [54].

The risk of bleeding is higher in older patients; the age group > 75 years shows the same pattern as younger patients with AF, that is, a reduction in the incidence of intracranial bleeding, and an increase in gastrointestinal bleeding when taking a DOAC compared with a VKA [55]. In RCTs compared with WAF, only API use was associated with a lower risk of major bleeding in patients > 75 years [56,57]. This group of patients is also inherently related to the problem of polypharmacy, which significantly affects the metabolism, especially of DOACs. Due to transport via the P-glycoprotein transport system, DOACs will be combined with many drugs used in the pharmacotherapy of patients suffering from cardiac diseases. Such drugs include amiodarone, dronedarone, diltiazem and finally digoxin [58,59]. For this reason, in patients treated with multiple drugs, special caution should be exercised when using anticoagulant therapy.

For comorbid liver disease, DOACs are contraindicated in patients with coagulopathy and clinically significant bleeding risk, including Child’s and Pugh’s class C cirrhosis, while RIV should not be used even in patients with AF and Child’s and Pugh’s class B cirrhosis because of the >2-fold increase in drug exposure in such patients [60]. DAB, API and EDO can be used with caution in patients with Child and Pugh class B cirrhosis. Lee et al. in a registry of cirrhotic patients treated primarily with low-dose DAB and RIV showed that the risk of stroke/systemic thromboembolism and also intracranial bleeding was comparable to WAF, and the risk of major bleeding, including gastrointestinal bleeding, was reduced [61].

A growing patient population and an ever-increasing problem in the context of anticoagulant treatment is occurring in patients with malignancies. The presence of malignant neoplasm has been shown to be associated with an increased risk of bleeding associated with thrombocytopenia, metastasis, renal and hepatic damage, vascular damage caused by tumor infiltration, invasive procedures and radiation therapy [62]. For patients with malignancies and AF, most of the data come from observational studies [63]. In RCTs involving patients with malignancies and venous thromboembolism, (Hokusai VTE Cancer trial with EDO and SELECT-D with RIV), it was shown that the use of DOACs compared with dalteparin was associated with a reduced rate of recurrent VTE and an increased risk of major bleeding, mainly from the gastrointestinal tract. The highest risk of bleeding was observed in patients with esophageal, gastric and urinary tract cancer [64,65]. However, it is worth noting that the situation is completely different when it comes to central nervous system (CNS) tumors. In 2022, a study was published in patients diagnosed with glioblastoma. In this group, the use of LMWH was associated with a significantly lower risk of intracranial hemorrhage compared to the use of DOACs [66]. However, data from another comparative study did not confirm this observation, stating that the risk is similar regardless of the type of anticoagulation treatment chosen [67]. In clinical practice, the use of DOACs in oncology patients shows considerable efficacy and safety, resulting in a significant benefit in patients with a good prognosis. However, the safety of their use in highly heterogeneous groups of patients with malignancies requires further studies [68,69].

## 5. Prevention of Bleeding Complications after Anticoagulation Therapy

In daily clinical practice, the decision regarding the type of anticoagulant therapy in individual cases is still based more on the physician’s subjective opinion than on objective evidence. This is due to the lack of clear data from clinical trials. The best way to prevent bleeding is to properly assess the bleeding risk before starting anticoagulation therapy. Since the release of the 2020 ESC guidelines, the most frequently tool used to assess this risk of bleeding is the HAS-BLED score [70]. However, it is not the only scale that can help in the assessment of this group of patients [71]. Table 3 presents the most frequently used scales for assessing the risk of bleeding and the risk factors that are taken into account in them.

Patients at higher risk of bleeding still pose the greatest challenge when making decisions about anticoagulant treatment. Especially since the risk of thromboembolic events in this group is often higher than in the general population [72]. In this situation, the best way to reduce the risk of bleeding seems to be to limit those modifiable risk factors (e.g., hypertension), which, according to the scales presented above, increase it [73]. Furthermore, it should be accented that the bleeding risk may change over time and the assessment should consider the dynamic changes.

Nowadays, when more and more specific reversal agents such as idarucizumab or andexenet-alfa are being used, it may seem that the use of these drugs for which we have a specific antidote is associated with greater safety for patients. However, it should be remembered that so far there are no data from RCTs that would indicate the superiority of using specific antidotes over supportive care, which is indicated in the case of bleeding after each anticoagulant [74].

## 6. Conclusions

Despite numerous comparative studies and meta-analyses conducted so far, there are still no clear data to determine which group of anticoagulants used in the prevention of thromboembolic events has a higher safety profile. Both their effectiveness and the risk of bleeding complications during their use are strongly dependent on the patient’s clinical profile, comorbidities and chronic medications used by the patient. For this reason, additional comparative studies are needed to better determine which anticoagulant therapy strategy will be beneficial in particular groups of patients. For this reason, the basic method of reducing the risk of bleeding in patients is still the reduction in those factors that increase this risk.

## Figures and Tables

**Figure 1 biomedicines-12-02242-f001:**
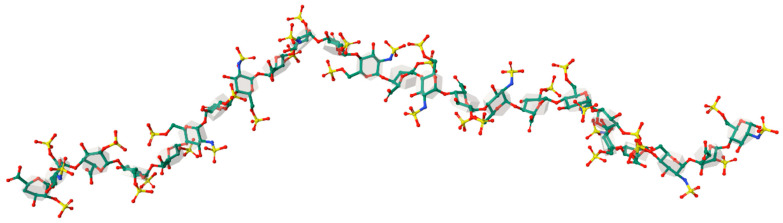
Visualization of a heparin polymer fragment. Visualization created based on the description of the crystal structure [13] using RSCB Protein DataBank 3D-viewer (RCSB.org, [14]).

**Figure 2 biomedicines-12-02242-f002:**
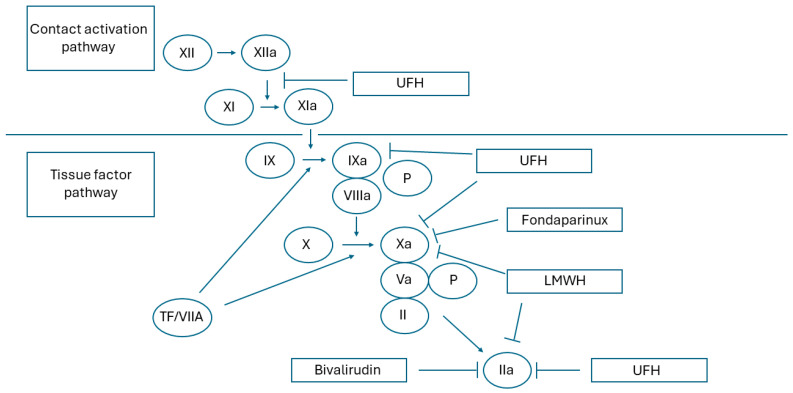
Mechanism of action for heparins. LMWH—low molecular weight heparin; P—phospholipid surface; TF—tissue factor; UFH—unfractionated heparin. Unfractionated heparin inhibits both the contact activation and tissue factor pathways. In contrast, LMWH and fondaparinux inhibit the tissue factor pathway only. Tissue factor pathway: tissue factor/factor VIIa complex (TF/VIIa) triggers the coagulation and activates factor IX (IXa) and factor X (Xa). Contact activation pathway: activation of factor XII (XIIa) initiates the coagulation and activates factor XI (XIa). Factor XIa activates factor IX and the activated factor IX (IXa) further results in the activation of factor X in a reaction that uses activated factor VIII (VIIIa) as a cofactor. Activated factor X (Xa) converts prothrombin (II) to thrombin (IIa) using activated factor V (Va) as a cofactor. Finally, thrombin converts fibrinogen to fibrin.

**Figure 3 biomedicines-12-02242-f003:**
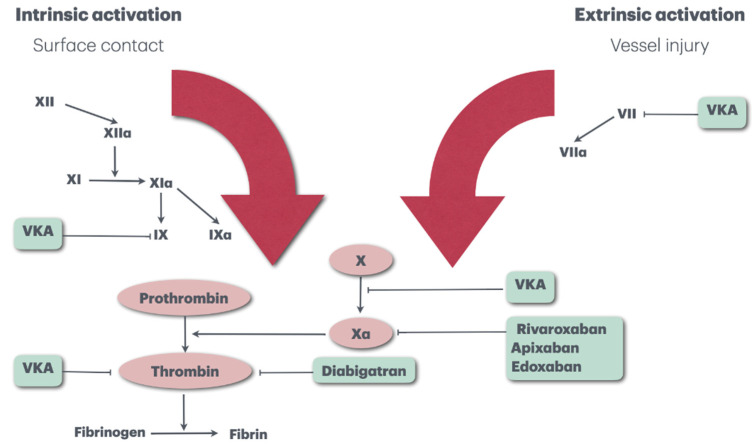
Site of action of DOACs and VKAs.

**Table 1 biomedicines-12-02242-t001:** Actions, pharmacokinetics and properties of DOACs [25,26].

Characteristics	Dabigatran	Riwaroxaban	Apixaban	Edoxaban
Mechanism of action	Direct inhibitor of thrombin	Factor Xa inhibitor	Factor Xa inhibitor	Factor Xa inhibitor
Bioavailability (%)	6	66 Served with meal: 80 to 100	50	62
Time to maximum concentration (h)	3	2 to 4	3	1 to 2
Serum half-life (h)	12 to 17	5 to 13	9 to 14	10 to 14
Dosage (mg)	2 × 110–150	1 × 15–20	2 × 2.5–5	1 × 30–60
Excretion by the kidneys (%)	80	33	27	50
Binding to proteins (%)	35	92–95	87	40–59
Membrane transporters	P-gp	P-gp, breast cancer immune protein	P-gp, breast cancer immune protein	P-gp
Hepatic CYP metabolism (%)	-	32	15	<4
Trials in atrial fibrillation	RE-LY	ROCKET-AF	ARISTOTLE AVERROES	ENGAGE-AF
Minimum permitted CrCL (mL/min)	30	15	15	15
Peculiarities	Dyspepsia in 5–10% of patients	Necessary to be served with a meal	-	-

**Table 2 biomedicines-12-02242-t002:** The Bleeding Academic Research Consortium (BARC) scale.

Type 0	No bleeding
Type 1	Bleeding that is not actionable and does not cause the patient to seek treatment
Type 2	Any clinically overt sign of hemorrhage that “is actionable” and requires diagnostic studies, hospitalization, or treatment by a health care professional
Type 3	a. Overt bleeding plus hemoglobin drop of 3 to <5 g/dL (provided hemoglobin drop is related to bleed); transfusion with overt bleeding b. Overt bleeding plus hemoglobin drop <5 g/dL (provided hemoglobin drop is related to bleed); cardiac tamponade; bleeding requiring surgical intervention for control; bleeding requiring IV vasoactive agents c. Intracranial hemorrhage confirmed by autopsy, imaging, or lumbar puncture; intraocular bleed compromising vision
Type 4	CABG-related bleeding within 48 h
Type 5	a. Probable fatal bleeding b. Definite fatal bleeding (overt or autopsy or imaging confirmation)

**Table 3 biomedicines-12-02242-t003:** Scores for estimating bleeding risk. BP—blood pressure; NSAIDs—nonsteroidal anti-inflammatory drugs; Hb—hemoglobin; Hct—hematocrit; hs-cTnT—high-sensitivity cardiac troponin T; GDF-15—growth differentiation factor 15; PPI—proton pump inhibitor.

Risk Score	Risk Factors
HAS-BLED	systolic BP > 160 mm Hg; severe renal or hepatic disease; stroke; previous bleeding; labile INR; age > 65; use of antiplatelets or NSAIDs; alcohol excess;
ABC	age; biomarkers (Hb, hs-cTnT, GDF-15 or cystatin C); previous bleeding
ATRIA	anemia; severe renal disease; age ≥ 75; previous bleeding; hypertension
Alfalfa-MB	age > 65; previous bleeding; anemia; vascular disease; no PPI; use of antiplatelets or NSAIDs; use of rivaroxaban
HEMORRHAGES	hepatic/renal disease; ethanol abuse; malignancy; age > 75; low platelets; re-bleeding risk; hypertension; anemia; genetic factors; increased falls risk; stroke
ORBIT	age ≥ 75; reduced Hb/Hct/anemia; previous bleeding; reduced renal function; use of antiplatelets

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
