# Peer review of "Bleeding Complications of Anticoagulation Therapy in Clinical Practice—Epidemiology and Management: Review of the Literature"

_biomedicines, 2024, doi:10.3390/biomedicines12102242_

Round 1

Reviewer 1 Report

Comments and Suggestions for Authors

The manuscript titled Bleeding complications of anticoagulation therapy in clinical 2 practice covers basic principals of bleeding complications that mostly arise due to various clinical practices. The review covers an important topic of bleeding complications and also covers the basic information of drugs used in this. 

A paragraph on prospective solutions could also be added with the conclusions. Also the title is only half. what is the message from the title is not fully clear?

Is this a full review or mini review. For full review 4-5 figures should be included with prospective solutions after conclusions. 

Reviewer 2 Report

Comments and Suggestions for Authors

An interesting and important topic is raised by the authors, since different types of anticoagulants are used in the clinical practice, however, little is known about their safety profiles under certain circumstances. After minor revision, the manuscript will be suitable for publication. Please find my comments, suggestions and questions below:

·        Please check the whole manuscript for typographical errors (i.e.: Table 1, correct typing of fondaparinux).

·        Section 1: I suggest to insert a schematic structural figures about UFH and LMWH

·        Figure 1: please re-edit figure 1 by discriminating between contact activation and tissue factor pathways and please label if certain parts of the coagulation cascade is blocked. Currently the figure only contains arrows, that makes it a bit confusing.

·        Section 1.2: Please introduce the abbreviations of vitamin K antagonists and novel oral anticoagulants. I also suggest to insert a figure about vitamin K antagonists’s mode of action.

·        Please fix the names of DOACs in Table 1.

·        Although it is described in Table 1 that DOACs are substrates of multidrug transporters, no information was given how to solve this issue and which anticoagulants can be used if the patient has active multidrug transporter. Please insert at least one sentence about it.

·        Section 1.3: Please add what kind of preventive actions can be taken to reduce the risk of bleeding.

Comments on the Quality of English Language

An interesting and important topic is raised by the authors, since different types of anticoagulants are used in the clinical practice, however, little is known about their safety profiles under certain circumstances. After minor revision, the manuscript will be suitable for publication. Please find my comments, suggestions and questions below:

·        Please check the whole manuscript for typographical errors (i.e.: Table 1, correct typing of fondaparinux).

·        Section 1: I suggest to insert a schematic structural figures about UFH and LMWH

·        Figure 1: please re-edit figure 1 by discriminating between contact activation and tissue factor pathways and please label if certain parts of the coagulation cascade is blocked. Currently the figure only contains arrows, that makes it a bit confusing.

·        Section 1.2: Please introduce the abbreviations of vitamin K antagonists and novel oral anticoagulants. I also suggest to insert a figure about vitamin K antagonists’s mode of action.

·        Please fix the names of DOACs in Table 1.

·        Although it is described in Table 1 that DOACs are substrates of multidrug transporters, no information was given how to solve this issue and which anticoagulants can be used if the patient has active multidrug transporter. Please insert at least one sentence about it.

·        Section 1.3: Please add what kind of preventive actions can be taken to reduce the risk of bleeding.
